

# Reliability of sonographic measurements of the ulnar collateral ligament: a multi-rater prospective study

Shawn D. Felton[1], Arie J. Van Duijn[2] and Mitchell L. Cordova[2]

[1] Marieb College of Health & Human Services Office of the Dean, Florida Gulf Coast University, Fort Myers, FL, United States of America
[2] Department of Rehabilitation Sciences, Florida Gulf Coast University, Fort Myers, FL, United States of America

## ABSTRACT

**Background.** The use of sonography is a cost-effective and reliable method to evaluate upper extremity superficial tissue structural integrity and pathology. Establishing the measurement reliability of widely used diagnostic ultrasound evaluation for musculoskeletal assessment is paramount enhance accurate clinical evaluations. The objective of this study was to establish the inter-rater and intra-rater reliability of select ulnar collateral ligament (UCL) thickness measures at two distinct anatomical locations in intercollegiate baseball athletes using ultrasound imaging (USI).

**Methods.** This was a prospective cohort study conducted in a university research laboratory and included a total of 17 NCAA Division I baseball athletes (age 20.4 $\pm$ 1.43, height = 183.63 cm $\pm$ 6.27 cm, mass = 89.28 kg $\pm$ 8.24 kg). Two trained clinicians measured UCL mid-substance and apex thickness in the throwing extremity, prospectively, on 5 occasions at 1-month intervals during rest. Intraclass correlation coefficients (ICCs) (model 3,3), associated standard error of measurement, and 95% minimal detectable change in thickness were derived.

**Results.** Intrarater reliability estimates for operator 1 were 0.90–0.98 (mid-substance) and 0.91–0.99 (apex). Operator 2's values were 0.92–0.97 and 0.93–0.99, respectively. The standard error of measurement (SEM) ranged from 0.045–0.071 cm (mid-substance) and 0.023–0.067 cm (apex). The minimal detectable difference (MDD95) was 0.12–0.20 cm (mid-substance) and 0.07–0.19 cm (apex). Interrater reliability was 0.86–0.96 (mid-substance) and 0.79–0.98 (apex); most ICCs were >0.90. Measurement of UCL thickness at two locations demonstrated very good to excellent reliability with high precision. Using this protocol, two evaluators can obtain consistent UCL measurement at two positions. This finding has significant implications for the clinical evaluation of superficial tissue pathology of the same individual by two experienced practitioners.

Corresponding author
Shawn D. Felton, sfelton@fgcu.edu

# INTRODUCTION

Efforts to prevent surgery intervention following injury to the ulnar collateral ligament (UCL) necessitates improvement in non-operative management. Accurate clinical decision

making is necessary for optimal conservative management which includes reliable and accurate assessment of the anatomical characteristics and mechanical integrity of the UCL. Ultrasound imaging (USI) has been used in medical practice since the 1950s. Recent reports note a sharp increase in the use of USI in clinical examination of musculoskeletal injury. This expansion is likely due to safety, portability, and relatively low-cost of USI compared to other imaging techniques (*Lento & Primack, 2008*; *Nazarian et al., 2003*; *Ciccotti et al., 2014*; *Henderson, Walker & Young, 2015*; *Jacobson et al., 2003*; *Wood, Konin & Nofsinger, 2010*). Furthermore, USI is an excellent alternative to other forms of imaging, particularly radiography, since all patients can safely undergo sonography as sonography transmits no ionizing radiation, is non-invasive, and provides a more patient-friendly experience by eliminating claustrophobia (*Ciccotti et al., 2014*; *Whittaker et al., 2007*). USI allows for an extremely precise and accurate dynamic, real-time evaluation of the underlying structures *in-vivo* (*Nazarian et al., 2003*).

Although the evaluation of musculoskeletal injuries can be greatly enhanced using USI (*Nazarian et al., 2003*), its utility is highly dependent upon the clinicians' skill proficiency, which is normally associated with steep learning curves (*Lento & Primack, 2008*). We posit that when used by trained healthcare professionals, especially point of care athletic trainers and physical therapists, employing proper techniques, USI represents an extremely valuable tool in further aiding in the diagnosis of elbow joint complex abnormalities (*Ferreira et al., 2015*; *Finlay, Ferri & Friedman, 2004*). The use of USI to evaluate medial elbow joint complex pain has been increasing in popularity due to its ability to enhance the diagnostic accuracy of the clinical exam. *Ciccotti et al. (2014)* showed that the use of stress USI was sufficient to detect structural changes to the UCL in asymptomatic professional baseball pitchers. Others have also demonstrated (*Nazarian et al., 2003*; *Roedl et al., 2016*) that when appropriately used conventional USI is as accurate as MRI arthrography in diagnosing UCL tears (*Nazarian et al., 2003*; *Roedl et al., 2016*).

Due the rising utility of using musculoskeletal ultrasound in the diagnosing the structural integrity of the UCL, it is essential that the measurements can derived with great reliability (*Nazarian et al., 2003*). A recent investigation has determined the reliability and precision of stress USI in measuring the length of the UCL and the joint gapping occurring at the medial joint space (*Bica et al., 2015*); however, this investigation only assessed joint gapping and not ligament morphology examined by a single clinician. Others have attempted to investigate the reliability of measuring the thickness of the common lateral extensors of the elbow where excellent intra-rater reliability and poor to good inter-rater reliability was observed (*Stewart et al., 2009*; *Toprak et al., 2012*). Other authors have studied the sonographic characteristics of the collateral ligaments of the elbow, including sonographic tissue signatures and thickness measurements of the UCL have typically been limited to mid-substance measurements only. These studies have reported mid-substance UCL thickening and hypoechoic changes in response to injury and increased loading. Increased valgus laxity was also reported, and no relationship was found between mid-substance thickness changes and valgus laxity. Thickness measurement of the UCL at the level of the trochlea may provide additional information regarding ligament integrity considering that this measurement is in closer anatomical proximity to the joint line,

however to date no reports in the literature were identified evaluating this measurement site (*Jacobson et al., 2003*; *Ferreira et al., 2015*; *Miller, Adler & Friedman, 2004*; *Sasaki et al., 2002*; *Teixeira et al., 2011*).

A range of studies have demonstrated the usefulness and effectiveness of USI in diagnosing orthopedic pathologies such as sprains, full thickness tears and localized swelling resulting from injuries, to date no investigations have examined the intra-rater and inter-rater reliability of UCL thickness over time and the large majority of prior studies involved highly trained physicians as raters. No studies have been found that examines how point of care providers such as athletic trainers and physical therapist could utilize USI in enhancing clinical diagnoses. Therefore, the purpose of this study was to determine the intra-rater and inter-rater reliability of clinically-relevant measurements of UCL at the ligament mid-substance and the apex of the trochlea with USI by point of care athletic trainers and physical therapists. Establishing the reliability of this measurement in this manner at two different locations will provide the critical foundation for which future clinical research seeking to better understand measuring UCL injury and its treatment, both non-invasively and *in vivo* can be conducted.

## MATERIALS & METHODS

A prospective cohort design guided this investigation where two primary outcomes were measured five times over the course of a single baseball season. These included the mid-substance and apex thickness of the UCL measure in centimeter (cm). The protocol used in this investigation was approved by the Florida Gulf Coast Institutional Review Board, study protocol 2015-51 and the informed consent was waived since the screening procedure was part of a pre-participation physical examination.

### Participants

We included a convenience sample of 17 male NCAA Division I collegiate baseball pitchers (age 20.4 $\pm$ 1.43 yrs, height = 183.63 $\pm$ 6.27 cm, mass = 89.28 $\pm$ 8.24 kg) from our university. Our subjects participated in this investigation if they had no prior history of UCL injury to their throwing extremity. Prior to participation in this study, all subjects gave written informed consent.

### Instruments

A GE Logiq e B12 point of care ultrasound system (GE Healthcare, Chicago, IL, USA) with a GE L4 12t linear transducer set at 10 MHz was used to provide real time long-axis imaging of the ulnar collateral ligament at two different anatomical locations with the skin prepped and cleaned with alcohol, then dried and a water-soluble gel applied prior to the examination. The ultrasound system uses digital beamforming with continuous dynamic receive focus and continuous dynamic receive aperture to optimize image resolution. The minimum depth of field ranges from 0–1 cm and the maximum depth of field ranges from 0–30 cm depending on the probe used. The display imaging depth also ranges between 0–30 cm. The GE L4-12t linear transducer operates at an imaging frequency between 4.2–13.0 MHz using a wide-band linear array.
### Study protocol

Brightness mode (B-mode) or 2-D mode USIs of the UCL of the throwing elbow were obtained by two trained sonographic practitioners using the GE Logiq e B12 ultrasound system (GE Healthcare, Chicago, IL, USA) for all images.

### Participant set-up

Participants were placed in supine position, with the elbow in 30 degrees flexion (as measured with standard goniometer), forearm supination, and shoulder in 60 degrees of abduction. This position was maintained throughout the image collection by a clinician. The proximal aspect of the transducer was placed on the medial epicondyle, with the distal aspect of the transducer on the medial aspect of the coronoid process of the proximal ulna (see Fig. 1), visualizing the anterior band of the UCL as described by *Jacobson et al. (2003)*.

### Operators

The two operators were both experienced in USI. The first operator was a licensed physical therapist with specialty certification as an orthopedic certified specialist with over 30 years of experience. The physical therapist operator had completed a continuing workshop of 24 h, and had completed over 250 musculoskeletal USI examinations. Furthermore, this operator has delivered numerous continuing education labs for other disciplines and professionals. The second operator was a licensed athletic trainer with over 20 years of experience. The athletic trainer had completed a similar continuing education workshop of 24 h and also received another four continuing education credits and has performed over 200 musculoskeletal USI examinations.

### Measurements

Long-axis sonographic imaging of the UCL of the throwing elbow were performed five times throughout the collegiate baseball season: start of the academic year, seven weeks later (prior to the start fall season), six weeks later (the end of fall season), nine weeks later (the beginning of the competition season), and 13 weeks later (the conclusion of the competition season). Each operator obtained three images of each participant, removing and repositioning the transducer between the acquisition of each image. All participants were evaluated on the same day in random order with 15 min between data collection and last throwing activity. Both operators performed measurements of the thickness of the UCL on all images after the data collection period had concluded. The thickness of the UCL was measured using the measurement function of the ultrasound system by placing a marker on the deep and superficial margins of the UCL. The thickness of the ligament was measured at the mid-substance at the deepest point of the fossa and at the level of the apex of trochlea (see Fig. 1), perpendicular to the ligament similar to the modified Jacobson-Ward technique described by *Shukla et al. (2017)*. In order to minimize inter-operator bias, operators were blinded to the identity of the participants and to the identity of the operator who performed the capture of the images during the measurement process by removing all personal identifiers including date of examination, athlete name, and operator identity from the image that was measured.

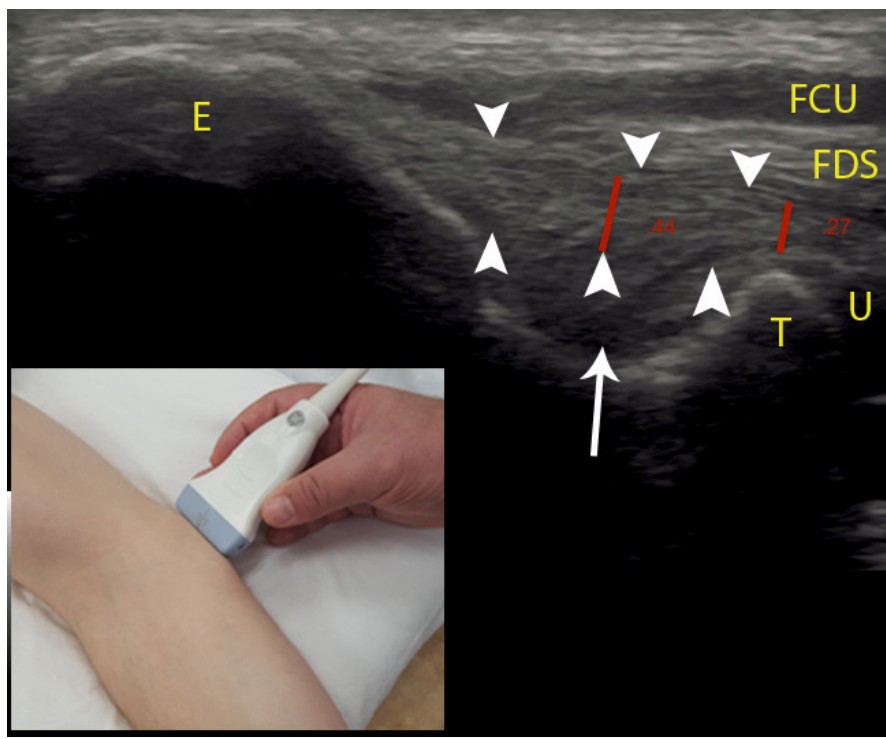

**Figure 1 Sonographic measurement of the UCL.** Positioning of the ultrasound probe is seen in the bottom left-hand corner of the image. Red lines denote UCL thickness measurements, arrowheads denote the UCL, arrow denotes subligamentous tissue. Abbreviations: E, medial epicondyle; T, trochlea; U, ulna; FDS, flexor digitorum superficialis; FCU, flexor carpi ulnaris.

## Data analysis

Statistical analysis was performed using SPSS version 26.0 (IBM Corp, Armonk, NY, USA). Descriptive statistics including the mean and standard deviation (SD) for each operator, measurement location, and session were calculated. Intrarater reliability for each operator was evaluated by calculating intraclass correlation coefficients ($ICC_{3,3}$) (*Shrout & Fleiss, 1979*) with 95% confidence intervals (CIs) for each measurement session, and measurement site. Measurement precision was evaluated by calculating the standard error of measurement (SEM) as:

$$SD * \sqrt{(1 - ICC)}$$ (*Portney & Watkins, 2020*).

The 95% minimal detectable difference (MDD95), representing the minimal difference in thickness that is needed to conclude with 95% confidence that a true change has occurred, was calculated as:

$$1.96 * SEM * \sqrt{(2)}$$ (*Shrout & Fleiss, 1979*).

Interrater reliability was evaluated by calculating ICCs ($model_{2,2}$) (*Shrout & Fleiss, 1979*) with 95%. CIs for each measurement session and measurement site, using the mean of three measurements by each operator on two different images obtained by each of the operators obtained at the same measurement session We used the interpretation guidelines described by *Portney & Watkins (2020)*: with excellent reliability defined as an ICC above

.90, good reliability above .75, moderate reliability between .50 and .75, and any values below .50 as poor.

## RESULTS

UCL thickness measurements (mean $\pm$ SD) for the two operators, intrarater $ICC_{3,3}$, SEM, and MDD95 are provided in Table 1. Mean thickness measurements at the UCL mid-substance by operator one ranged from 0.49–0.55 cm, and from 0.51–0.56 cm for operator two. Mean thickness measurements at the apex of the trochlea by operator one ranged from 0.31–0.37 cm, and from 0.32–0.35 cm for operator two. Intrarater reliability estimates for operator one as expressed by $ICC_{3,3}$ ranged from 0.90–0.98 at the mid-substance, and 0.91–0.99 at the apex of the trochlea. Operator two $ICC_{3,3}$ scores ranged from 0.92–0.97 at the mid-substance and 0.93–0.99 for the apex of the trochlea, respectively. Measurement precision, as expressed by the SEM, ranged from 0.045–0.071 cm at the mid-substance, and 0.023–0.067 cm at the apex of the trochlea. The MDD95 ranged from 0.12–0.20 cm at the mid-substance and 0.07–0.19 cm at the apex. Interrater reliability ranged from 0.86–0.96 at the UCL mid-substance, and from 0.79–0.98 at the apex of the trochlea. A large majority of ICCs did exceed 0.90 (see Table 2).

## DISCUSSION

Comparison of two USI operators demonstrated excellent and good to excellent reliability and low SEM values for measuring the thickness of the UCL at the mid-substance of and apex of the UCL respectively using sonography in intercollegiate baseball athletes. Our method of measuring the thickness of the UCL can be used to provide a reliable estimate of ligament thickness that can be used by clinicians in assessing the structural integrity in their patients. Demonstrating consistent measurement of these structures, as indicated by the low SEMs, is valuable in non-surgical treatment and management of this common injury in baseball athletes. The intra- and interrater reliability was similar regardless of measurement date and operator who attained the image. Our results are similar to the results found by *Shukla et al. (2017)* and *Ferreira et al. (2015)* for interrater reliability, and exceeded the ICC values of .67 reported *Keller et al. (2015)*. However, what further differentiates this study was the analysis of intrarater reliability among point of care clinicians. The apparent lower inter-rater ICC scores were plausibly a result of slightly different placement of the US transducer by the two examiners while conducting the exam.

The reliability of the trochlea measurements was slightly higher than mid-substance measurements of the UCL thickness. A possible explanation for this is that the UCL directly overlies the bony trochlea, potentially making it easier for the operator to identify the deep ligament margin at this anatomical location. The deep mid-substance margin is less easy to identify due to the adjacent soft tissue with tissue signature that is less distinct from the ligament compared to the bony tissue at the trochlea location (see Fig. 1). Thickness measurement of the UCL at the level of the trochlea may provide additional information regarding ligament integrity considering that this measurement is in closer anatomical proximity to the joint line.

**Table 1  UCL thickness measurements and intrarater reliability estimates.**

| Measurement | Mean (SD) in cm | ICC$_{3,3}$; (95% CI) | SEM | MDD$_{95}$ |
|---|---|---|---|---|
| Operator 1 | | | | |
| Measurement date 1 | | | | |
|     Mid-substance | 0.49 (0.11) | 0.98 (0.96–0.99) | 0.045 | 0.12 |
|     Trochlea | 0.37 (0.09) | 0.97 (0.94–0.99) | 0.052 | 0.14 |
| Measurement date 2 | | | | |
|     Mid-substance | 0.51 (0.07) | 0.90 (0.79–0.96) | 0.067 | 0.19 |
|     Trochlea | 0.34 (0.07) | 0.91 (0.82–0.97) | 0.059 | 0.16 |
| Measurement date 3 | | | | |
|     Mid-substance | 0.52 (0.06) | 0.91 (0.81–0.97) | 0.056 | 0.15 |
|     Trochlea | 0.34 (0.08) | 0.98 (0.95–0.99) | 0.035 | 0.09 |
| Measurement date 4 | | | | |
|     Mid-substance | 0.54 (0.07) | 0.94 (0.87–0.98) | 0.050 | 0.14 |
|     Trochlea | 0.35 (0.08) | 0.99 (0.98–0.99) | 0.023 | 0.07 |
| Measurement date 5 | | | | |
|     Mid-substance | 0.54 (0.06) | 0.91 (0.81–0.97) | 0.053 | 0.15 |
|     Trochlea | 0.31 (0.09) | 0.97 (0.94–0.99) | 0.043 | 0.12 |
| Operator 2 | | | | |
| Measurement date 1 | | | | |
|     Mid-substance | 0.51 (0.11) | 0.97 (0.93–0.99) | 0.060 | 0.17 |
|     Trochlea | 0.35 (0.13) | 0.98 (0.96–0.99) | 0.050 | 0.14 |
| Measurement date 2 | | | | |
|     Mid-substance | 0.51 (0.08) | 0.92 (0.83–0.97) | 0.071 | 0.20 |
|     Trochlea | 0.33 (0.07) | 0.95 (0.89–0.98) | 0.045 | 0.13 |
| Measurement date 3 | | | | |
|     Mid-substance | 0.54 (0.08) | 0.93 (0.83–0.97) | 0.069 | 0.19 |
|     Trochlea | 0.32 (0.09) | 0.99 (0.97–0.99) | 0.030 | 0.08 |
| Measurement date 4 | | | | |
|     Mid-substance | 0.56 (0.06) | 0.91 (0.80–0.96) | 0.060 | 0.17 |
|     Trochlea | 0.34 (0.07) | 0.98 (0.96–0.99) | 0.031 | 0.09 |
| Measurement date 5 | | | | |
|     Mid-substance | 0.54 (0.07) | 0.92 (0.83–0.97) | 0.061 | 0.17 |
|     Trochlea | 0.34 (0.09) | 0.93 (0.85–0.97) | 0.069 | 0.19 |

**Notes.**

Abbreviations: ICC, intraclass correlation coefficient; SEM, standard error of measurement; MDC, minimal detectable difference.

Values in parentheses are 95% confidence interval.

The ability of clinicians to reliably measure the thickness of the UCL can contribute to the clinical decision making regarding potential UCL dysfunction and pathology. Various authors have reported UCL thickening and valgus laxity in response to increased loading (*Jacobson et al., 2003*; *Ferreira et al., 2015*; *Miller, Adler & Friedman, 2004*; *Sasaki et al., 2002*; *Teixeira et al., 2011*; *Keller et al., 2015*). *Shukla et al. (2017)* describe how early identification of these UCL changes may be useful in preventing future injury.

**Table 2  Interrater reliability estimates for each anatomical location and day of exam.**

| Measurement | $ICC_{2,2}$; (95% CI) |
| --- | --- |
| Measurement date 1 | |
|    Mid-substance Image 1 | 0.90 (0.74–0.97) |
|    Image 2 | 0.92 (0.79–0.97) |
|    Trochlea Image 1 | 0.98 (0.93–0.99) |
|    Image 2 | 0.98 (0.94–0.99) |
| Measurement date 2 | |
|    Mid-substance Image 1 | 0.96 (0.89–0.99) |
|    Image 2 | 0.93 (0.81–0.98) |
|    Trochlea Image 1 | 0.94 (0.83–0.98) |
|    Image 2 | 0.79 (0.42–0.93) |
| Measurement date 3 | |
|    Mid-substance Image 1 | 0.92 (0.78–0.97) |
|    Image 2 | 0.89 (0.67–0.96) |
|    Trochlea Image 1 | 0.98 (0.94–0.99) |
|    Image 2 | 0.96 (0.88–0.99) |
| Measurement date 4 | |
|    Mid-substance Image 1 | 0.90 (0.70–0.96) |
|    Image 2 | 0.94 (0.83–0.98) |
|    Trochlea Image 1 | 0.94 (0.82–0.98) |
|    Image 2 | 0.98 (0.94–0.99) |
| Measurement date 5 | |
|    Mid-substance Image 1 | 0.90 (0.70–0.96) |
|    Image 2 | 0.86 (0.58–0.95) |
|    Trochlea Image 1 | 0.91 (0.74–0.97) |
|    Image 2 | 0.96 (0.88–0.99) |

**Notes.**
Abbreviations: ICC, intraclass correlation coefficient.
Values in parentheses are 95% confidence interval.

Limitations of this study include a relatively small sample size that was homogeneous in terms of sex and activity level. The participants in this study were asymptomatic, and since sonographic characteristics of injured *vs* non-injured tissue may differ, this study should be repeated in a symptomatic population. The study was completed over the course of a single baseball season, and thus the measurement dates were relatively close to each other. This increased the potential that the operators gained familiarity with the participants, however the results of this study did not show an increase in reliability over measurement dates.

## CONCLUSIONS

The method of measuring UCL thickness using sonography at two different anatomical locations described in this study was shown to have good to excellent reliability. This measurement method may be used in future research involving measurements of the thickness of the UCL by utilizing the reported values in this study as a reference value for identifying UCL abnormalities caused by baseball related injuries in pitchers. The method described in this study is feasible both for research and clinical applications for assessing superficial soft tissues of the elbow and may have utility in the assessment of other peripheral joints.

The method of measuring UCL thickness using sonography at two different anatomical locations described in this study was shown to have good to excellent reliability.

Measurements of the thickness of the UCL can be obtained by two different clinicians over the course of a baseball season following the same protocol. These data have significant positive implications in identifying UCL pathology in the throwing athlete prospectively with the potential for predicting injury risk of these structures. More research is needed to expand this protocol for measuring other superficial soft tissue structures with large clinical manifestations such as the collateral ligaments of the knee joint complex.

### Funding

The authors received no funding for this work.

### Competing Interests

The authors declare there are no competing interests.

### Author Contributions

- Shawn D. Felton conceived and designed the experiments, performed the experiments, analyzed the data, prepared figures and/or tables, authored or reviewed drafts of the article, and approved the final draft.
- Arie J. Van Duijn conceived and designed the experiments, performed the experiments, analyzed the data, prepared figures and/or tables, authored or reviewed drafts of the article, and approved the final draft.
- Mitchell L. Cordova performed the experiments, analyzed the data, authored or reviewed drafts of the article, and approved the final draft.

### Human Ethics

The following information was supplied relating to ethical approvals (*i.e.*, approving body and any reference numbers):

Florida Gulf Coast University

### Data Availability

The raw measurements are available in the Supplementary File.

## Supplemental Information

Supplemental information for this article can be found online at http://dx.doi.org/10.7717/peerj.15418#supplemental-information.

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
