# Peer review of "Reliability of sonographic measurements of the ulnar collateral ligament: a multi-rater prospective study"

_PeerJ, doi:10.7717/peerj.15418_

## Round 0.1 · original submission · Major Revisions

According to the analysis carried out by this editor and the reviewers, the manuscript needs greater depth in the introduction and research methods. In this context, the authors need to address the existing literature on sonographic measurements of the ulnar collateral ligament, as well as provide more details on the procedures and assessments applied in the research. These aspects impact the validity of the findings. For more details, please review the reviewers' comments.

Reviewer 1 ·

Basic reporting

No comment

Experimental design

Most of my comments are targeted at the procedures presented by the authors, which should be described in greater detail.

Validity of the findings

The validity of the results depends in part on the clarity of the procedures. It is for this reason that the conclusions should be rechecked by the authors.

Annotated reviews are not available for download in order to protect the identity of reviewers who chose to remain anonymous.

·

Basic reporting

The writing is mostly easy to follow and succinctly describe the purpose, experimental design, and results of the study. The discussion is clear as well. Specific comments on a few small issues are below:

- Lines 56-58: This statement is hard to follow. My suggestion would be to split the sentences and delete the comment about orthopedic setting to state something like “Recent reports note a sharp increase in the use of USI in clinical examination of musculoskeletal injury. This expansion is likely due to the safety, portability, and relatively low-cost of USI compared to other imaging techniques.”

- Line 63: The sentence starting with “Although the evaluation…” should start another paragraph, as it’s a different idea than the introduction that comes before it.

- Line 75-77: Seems like a run-on sentence. Dividing this point into 2 sentences would be advantageous. Also, the authors could eliminate the word “However” from the beginning of the sentence. This sentence also seems to be out of place. Moving it to the end of the introduction to combine with the sentence starting on line 94 (“Whereas a range…”)

- Line 84: language in parentheses not needed here

- Line 94: Moving the sentence starting “Whereas a range…” into a different paragraph would help clarity.

- Line 117: Words are “were used” at the end of the sentence are not needed

- Line 192-195: This sentence doesn’t make sense and needs to be edited

- Line 202: The discussion of differences in the two measurement techniques and thickness over time should be in a separate paragraph for clarity. For further comments on this section, please see “Validity of the Findings Section.”

Experimental design

The authors do a great job of establishing the need for this work. Experimental design well described and able to be replicated. Figure 1 does a great job in identifying measurement technique.

Validity of the findings

The results and discussion mostly match the research question. Only two small issues exist, one in the data set and one in the discussion. More complete comments on those issues are below.

- In the data set, participant 15 has 999 values from 11/13 to the end of the study. My assumption would be that this participant either became injured and/or left the team and that his data were not included in the later ICC values. This participant should either be deleted from the study entirely, or his dropout should be noted in the results. If the participant is still included, the authors should be clear about what data was included for that participant.

- Line 202-204: While this is an important finding, it doesn’t quite fit the overarching purpose of the article. Instead of speculating on the trend of thickening over time, the authors should note that the reliability of the trochlea measurements were slightly higher than mid-substance. two measurements (trochlea and mid-substance first) and then speculate on the reasons why. In this discussion, the authors should consider citing Keller et all. 2015- Pre- and Postseason Dynamic Ultrasound Evaluation of the Pitching Elbow.

Additional comments

The authors should be commended on their work. The study is clear and concise. It also fits the need in the literature to establish baseline values and show reliability of USI as it becomes more relevant in physical therapy clinics and athletic training rooms.

·

Basic reporting

Reviewer comments:
Introduction:
Well written, however I have general comments:
Line 53-56, 66, 79, 88-98: Need references.
Line 79-83 can be merged into a single sentence

Materials and methods:
This section needs work:
Line 130: Id prefer a section called “Examiners, raters, operator” and we need stringency in terminology, e.g. decide whether you use operator or rater! Is the operator the same as the rater?
And then the section “Study protocol” should be divided into sub-sections for clarity: participant setup, raters, measurements etc.
It must be clear to the reader what was performed live with the participant and what was measured subsequently.

Discussion:
Rather short discussion that really needs more reflection in relation to other studies and clinical reasoning, especially with the focus of the purpose.
Further, lack references e.g. line 202-204.

Experimental design

Make sure you cover the existing literature in the background.

Validity of the findings

Please provide more reflections in relation to other studies and clinical reasoning, especially with the focus of the purpose.

---

## Round 0.2 · accepted · Accept

According to the analysis carried out by this editor and the reviewers, the manuscript has the necessary elements and merits to be accepted.

Reviewer 1 ·

Basic reporting

no comment

Experimental design

no comment

Validity of the findings

no comment

·

Basic reporting

No comment

Experimental design

No comment

Validity of the findings

No comment

Additional comments

No comment